# Relationship between age at menarche and metabolic diseases in Korean postmenopausal women: The Korea National Health and Nutrition Examination Survey 2016–2018

**Hyein Jung(iD), Yeon-Ah Sung, Young Sun Hong, Do Kyeong Song, So-hyeon Hong, Hyejin Lee(iD)***

Division of Endocrinology and Metabolism, Department of Internal Medicine, Ewha Womans University College of Medicine, Seoul, South Korea

* hyejinlee@ewha.ac.kr

**Data Availability Statement:** All relevant data are within the paper.

## Abstract

### Background

Cardiovascular disease is the leading cause of morbidity and mortality in postmenopausal women. Early menarche may be associated with an increased risk of metabolic diseases such as diabetes and cardiovascular disease. This study aimed to investigate the effect of menarche age and the risk of diabetes and metabolic syndrome in Korean postmenopausal women.

### Methods

We analyzed 4,933 postmenopausal women (mean age: 64.7 years) using the Korean National Health and Nutritional Examination Survey 2016–2018. Subjects were divided into three groups according to menarche age (early menarche: $\leq$ 12 years (n = 451), reference: 13–16 years (n = 3,421), and late menarche: $\geq$ 17 years (n = 1,061)). Logistic regression analysis was used to estimate the odds ratio (OR) for diabetes and metabolic syndrome.

### Results

Women with an early menarche age were younger, more educated, and had higher income than the other groups (p-value < 0.001). There were no differences in body mass index, blood pressure, fasting glucose, HbA1c, and cholesterol levels among the three groups. After adjusting for potential confounding factors, early menarche age was significantly associated with the risk of diabetes (OR 1.435, 95% confidence interval (CI): 1.069–1.928). The prevalence of metabolic syndrome in all subjects was 41.1%. After adjusting for potential confounding factors, the OR of metabolic syndrome in the early menarche group was 1.213 (95% CI: 0.971–1.515)

**Funding:** The authors received no specific funding for this work.

**Competing interests:** The authors have declared that no competing interests exist.

## Conclusion

The risk of diabetes was 1.43 times higher in postmenopausal Korean women with early menarche. Although the risk of metabolic syndrome was not statistically significant, it showed a tendency to increase in the early menarche group. Our results suggest that age at menarche may be helpful in diabetes risk stratification and early interventions for postmenopausal women.

## Introduction

Diabetes affects 400 million people worldwide, and the disease and its complications pose a high burden on public health [1]. In Korea, 13.8% of adults aged $\geq$ 30 years have diabetes, and the prevalence of diabetes have increased [2]. Metabolic syndrome(MetS) is a clustering of at least three of the following medical conditions: central obesity, hyperglycemia, dyslipidemia, and hypertension [3]. Diabetes and MetS are more common in the older population and are directly related to cardiovascular events [4, 5]. Therefore, it is important to identify those at high risk of diabetes and MetS.

Age at menarche is defined as the age at which menstruation begins and is one marker of puberty. Age at menarche is influenced by several factors, including genetics, body weight, ethnicity, socioeconomic status, and nutrition [6, 7]. Several studies have demonstrated that early menarche increases the risk of diabetes and cardiovascular disease [8–10]. Most studies have been conducted in Western countries, although similar results were found in a recent study on Chinese postmenopausal women [11]. There have been several studies on the association between MetS and age at menarche, but the results have been inconsistent. Epidemiologic studies showed positive associations between early menarche and prevalence of MetS [12–14]; however, in some works, the association was not statistically significant [15]. Also, most of the research was conducted on Western populations.

In this study, we investigated whether age at menarche was associated with the risk of diabetes and MetS in Korean postmenopausal women from national survey data from the Korea National Health and Nutrition Examination Survey (KNHANES) 2016–2018.

## Methods

### Study population

We used national survey data of the KNHANES VI (2016–2018). The KNHANES is a cross-sectional and nationwide survey that collects data on the health status and nutritional intake of Koreans annually. We included the postmenopausal women of the 24,269 participants in KNHANES 2016–2018. Premenopausal women were excluded. Participants answered the following questions: "Are you on a menstruation period currently?" or "When was your age at menopause?" Women under the age of 45 were excluded to minimize the number of women with artificial menopause. Women with an onset of age at menarche under 10 years and over 19 years were excluded. We also excluded participants with missing responses for data such as age at menarche. Finally, we analyzed 4,933 women. The KNHANES was approved by the Institutional Review Board of the Korea Centers for Disease Control and Prevention. All study participants provided written informed consent. The data is available to the public on the KNHANES website.

## Definition of variables

We used demographic variables (age and socioeconomic status), lifestyle factors (smoking, alcohol drinking, and physical activity), and reproductive factors (use of oral contraceptive pills, age at menopause, and menopause status) from KNHANES data. Socioeconomic characteristics included income and education level. Income was divided into quartiles (lowest, lower-middle, upper-middle, and highest). Education level was divided into quartiles ($\leq$ elementary school, $\leq$ middle school, $\leq$ high school, and $\geq$ university) Current smokers were defined as persons who smoked more than five packs in their lifetime and who smoked currently. Alcohol use was defined as drinking more than one drink per month over the past year.

Bodyweight and height measurements were measured by an experienced person. Body mass index (BMI) was calculated as weight (kg)/height squared ($m^2$). Waist circumference (WC) was measured at the area between the rib cage and the iliac crest. Blood pressure was measured three times by a mercury sphygmomanometer on the right arm and assessed as the average of the second and the third blood pressure measurements. Blood samples were collected after fasting for more than eight hours. Serum levels of fasting blood glucose, Hemoglobin A1c (HbA1c), total cholesterol (TC), high-density lipoprotein (HDL) cholesterol, and triglyceride (TG) were measured.

## Assessment of age at menarche

Age at menarche was defined as the age at the first menstruation. The information was obtained by self-reporting using a standard questionnaire: "When did you have your first menstrual period?" We categorized the age at menarche into three groups: $\leq$ 12 years, 13–16 years (reference group), and $\geq$ 17years

## Definition of diabetes and metabolic syndrome

Diabetes was assessed through a questionnaire. Subjects were asked if they had ever been diagnosed with either type 1 or type 2 diabetes.

MetS was defined according to the Internal Diabetes Federation (IDF) [16] as any three or more of the followings: 1) waist circumference for Korean women $\geq$ 80 cm, 2) fasting triglycerides level $\geq$ 150 mg/dL, 3) HDL cholesterol < 50 mg/dL, 4) systolic blood pressure $\geq$ 130 mmHg, or a diastolic BP $\geq$ 85 mmHg, or taking medication for high blood pressure, and 5) fasting plasma glucose $\geq$ 100 mg/dL or taking medication to treat diabetes.

## Statistical analysis

SPSS Statistics version 25 (IBM Corp., Armonk, NY, USA) was used for statistical analyses. P-values less than 0.05 indicated statistical significance. Continuous variables were expressed as mean ± standard deviation. Categorical variables were expressed as counts and percentages (%). One-way ANOVA and the Chi-square test were used to determine the statistical significance of continuous and categorical variables, respectively.

Multivariate binary logistic regression analysis was performed to estimate the risk of diabetes and MetS according to age at menarche. Model 1 was not adjusted. Model 2 was adjusted for age at recruitment. Model 3 was further adjusted for education level and income. Model 4 was additionally adjusted for lifestyle factors (smoking, alcohol consumption, and physical activity levels).

# Results

## Baseline characteristics of the participants

The baseline characteristics of the participants are presented in Table 1. The mean age was 64.7 years. Overall the mean menarche age was 14.9 years and the mean age at menopause onset was 49.2 years. Women with an early menarche age were younger, had a higher income, and were more educated than women in the other groups (p < 0.001). They were likely to have smoked or consumed alcohol. There was no significant difference in the prevalence of hypertension, diabetes, and dyslipidemia among the three groups.

## Metabolic parameters

The metabolic parameters of the participants are shown in Table 2. The average BMI was 24.3 and waist circumference was 82.3 cm. There was no difference in BMI between the three groups and waist circumference. After age adjustment, there was no difference between the three groups in systolic/diastolic blood pressure, fasting blood glucose, HbA1c, total cholesterol, and TG levels. Only HDL cholesterol was significantly higher in the early menarche age group.

## Risk of diabetes by age at menarche

The relationship between age at menarche with diabetes is shown in Table 3. Before adjustment, the late menarche group had a high odds ratio (OR) for diabetes. (OR 1.348, 95% CI:

**Table 1. Baseline characteristics of study participants by age at menarche.**

| | total | Age at menarche | | | p-value |
|---|---|---|---|---|---|
| | | early menarche (<13) | reference (13–16) | late menarche ≥17) | |
| Number of participants | 4933 | 451 (9.1%) | 3421 (69.3%) | 1061 (21.5%) | |
| Age, years | 64.7 ± 9.1 | 59.5 ± 8.1 | 63.9 ± 8.9 | 69.4 ± 8.2 | <0.001 |
| age at menarche | 14.9 ± 1.9 | 11.8 ± 0.5 | 14.6 ± 1.0 | 17.5 ± 0.7 | <0.001 |
| Age at menopause | 49.3 ± 4.9 | 48.8 ± 4.8 | 49.4 ± 4.7 | 49.1 ± 5.7 | 0.061* |
| Income | | | | | <0.001 |
| Lowest | 1196 (24.3%) | 98 (21.7%) | 800 (23.5%) | 298 (28.2%) | |
| Lower middle | 1250 (25.4%) | 120 (26.6%) | 835 (24.5%) | 295 (27.9%) | |
| Higher middle | 1222 (24.9%) | 95 (21.1%) | 873 (25.6%) | 254 (24.0%) | |
| Highest | 1246 (25.4%) | 138 (30.6%) | 898 (26.4%) | 210 (19.9%) | |
| Education | | | | | <0.001 |
| Elementary school | 2302 (46.7%) | 117 (25.9%) | 1429 (41.8%) | 756 (71.3%) | |
| Middle school | 824 (16.7%) | 53 (11.8%) | 600 (17.6%) | 171 (16.1%) | |
| High school | 1191 (24.2%) | 161 (35.7%) | 924 (27.0%) | 106 (10.0%) | |
| University | 613 (12.4%) | 120 (26.6%) | 465 (13.6%) | 28 (2.6%) | |
| Current smoker | 321 (6.5%) | 43 (9.5%) | 216 (6.3%) | 62 (5.8%) | 0.115* |
| Alcohol | 1369 (27.8%) | 155 (34.4%) | 991 (29.0%) | 223 (21.2%) | 0.592* |
| Physically active | 1698 (34.6%) | 202 (44.9%) | 1197 (35.1%) | 299 (28.3%) | 0.024* |
| Ever use of oral contraceptives | 1109 (22.5%) | 77 (17.1%) | 801 (23.4%) | 231 (21.8%) | 0.010 |
| Disease diagnosis | | | | | |
| Hypertension | 2028 (41.1%) | 141 (31.3%) | 1375 (40.2%) | 512 (48.3%) | 0.343* |
| Diabetes mellitus | 757 (15.3%) | 64 (14.2%) | 493 (14.4%) | 200 (18.9%) | 0.154* |
| Dyslipidemia | 1754 (35.6%) | 145 (32.2%) | 1248 (36.5%) | 361 (34.0%) | 0.002* |

* Age adjusted p-value

**Table 2. Metabolic parameters of study participants by age at menarche.**

| | total | Age at menarche | | | P-value age adjust |
|---|---|---|---|---|---|
| | | early menarche (<13) | reference (13–16) | late menarche (≥17) | |
| BMI, kg/m2 | 24.3 ± 3.4 | 24.5 ± 0.2 | 24.3 ± 0.0 | 24.2 ±0.1 | 0.201 |
| Waist circumference, cm | 82.3 ± 9.1 | 82.7 ± 0.4 | 82.31 ± 0.15 | 82.1 ± 0.3 | 0.511 |
| Systolic blood pressure (mmHg) | 125.7 ± 18.7 | 125.2 ± 0.8 | 125.1 ± 0.3 | 125.0 ± 0.6 | 0.981 |
| Diastolic blood pressure(mmHg) | 74.6 ± 9.9 | 74.4 ± 0.5 | 74.5 ± 0.2 | 75.0 ± 0.3 | 0.352 |
| Fasting glucose (mg/dL) | 104.1 ± 25.2 | 104.3 ± 1.2 | 104.2 ± 0.4 | 103.5 ± 0.8 | 0.746 |
| HbA1c (%) | 5.9 ± 0.8 | 5.9 ± 0.0 | 5.9 ± 0.0 | 5.9 ± 0.0 | 0.971 |
| Total cholesterol (mg/dL) | 198.4 ± 40.6 | 197.9 ± 1.9 | 198.1 ± 0.7 | 199.7 ± 1.3 | 0.545 |
| Triglyceride (mg/dL) | 131.3 ± 91.7 | 129.0 ± 4.5 | 130.8 ±1.6 | 133.8 ± 3.0 | 0.599 |
| HDL cholesterol (mg/dL) | 52.1 ± 12.6 | 53.3 ± 0.6 | 52.2 ± 0.2 | 51.3 ± 0.4 | 0.024 |
| Metabolic syndrome (%) | 1924 (41.1%) | 156 (36.2%) | 1294 (39.6%) | 474 (48.3%) | 0.519 |

1.124–1.616) The OR in women with an early menarche age was 0.978. After adjustment for several variables, the OR in women with early menarche age increased to 1.435 (95% CI: 1.069–1.928). The OR in women with late menarche age was not statistically significant (OR 0.916, 95% CI: 0.755–1.113).

## Risk of metabolic syndrome

In total, 1,924 (41.1%) subjects met the MetS criteria. The prevalence of MetS increased according to menarche age (36.2% in the early menarche group, 39.6% in the reference group, and 48.3% in the late menarche group). The age-adjusted p-value was 0.519 (Table 2).

The OR for MetS are summarized in Table 4. Before adjustment, the late menarche group had high OR for MetS (OR 1.425, 95% CI: 1.234–1.645). After adjusting for several variables, the OR in women in the early menarche age group increased to 1.213 (95% CI: 0.971–1.515), but this was not significant statistically.

## Discussion

In this study, early menarche was associated with the risk of diabetes in postmenopausal Korean women. Although the risk of MetS was not statistically significant, it showed a tendency to increase in the early menarche group. There were no differences in body mass index, blood pressure, fasting glucose, HbA1c, and cholesterol level among the three groups.

Our findings of the positive relationship between early menarche and diabetes are consistent with previous studies. Several large prospective cohort studies have demonstrated that

**Table 3. Age at menarche and risk of diabetes mellitus.**

| | Age at menarche | | |
|---|---|---|---|
| | early menarche (<13) | reference (13–16) | late menarche ≥17 |
| | OR (95% CI) | | OR (95% CI) |
| Model 1 | 0.978 (0.738–1.295) | 1.00 | 1.348 (1.124–1.616) |
| Model 2 | 1.318 (0.986–1.763) | 1.00 | 0.975 (0.807–1.180) |
| Model 3 | 1.404 (1.046–1.884) | 1.00 | 0.938 (0.774–1.136) |
| Model 4 | 1.435 (1.069–1.928) | 1.00 | 0.916 (0.755–1.113) |

Model 1: unadjusted. Model 2: adjusted for age. Model 3: adjusted for age, education and income level. Model 4: adjusted for age, education, income level, smoking, alcohol intake and physical activity.

**Table 4. Age at menarche and risk of metabolic syndrome.**

| | Age at menarche | | |
| --- | --- | --- | --- |
| | early menarche (<13) | reference (13–16) | late menarche ≥17 |
| | OR (95% CI) | | OR (95% CI) |
| Model 1 | 0.866 (0.703–1.067) | 1.00 | 1.425 (1.234–1.645) |
| Model 2 | 1.120 (0.902–1.390) | 1.00 | 1.051 (0.903–1.223) |
| Model 3 | 1.195 (0.958–1.490) | 1.00 | 0.955 (0.818–1.114) |
| Model 4 | 1.213 (0.971–1.515) | 1.00 | 0.958 (0.820–1.120) |

Model 1: unadjusted. Model 2: adjusted for age. Model 3: adjusted for age, education and income level. Model 4: adjusted for age, education, income level, smoking, alcohol intake and physical activity

early menarche is associated with an increased risk of diabetes in adulthood [8, 9]. Women in the earliest menarche group had a 70% higher incidence of type 2 diabetes in the EPIC-Inter-Act study, and less than half of this association appears to be mediated by a higher adult BMI [8]. A recent study of postmenopausal women in a Chinese rural cohort showed that the risk of type 2 diabetes decreased by 6% as the menarche age was delayed by one year [11]. Although the age of menarche was higher in Asian women than in Western women, the association between menarche age and diabetes showed the same trend.

The relationship between menarche age and MetS was inconsistent across studies. A systematic review and meta-analysis of 16 studies suggested that early menarche is associated with a greater risk of MetS (pooled RR: 1.62, 95% CI: 1.40–1.88) [17]. In contrast, based on KNHANES 2005, Cho et al. suggest that menarche age was not associated with MetS [18]. A meta-analysis based on nine studies of both pre- and postmenopausal women showed that there was no relationship between menarche age and overall cardiovascular death [19]. In this study, we observed a trend for increased risk of MetS with decreasing menarche age, which was not statistically significant. These differences may have been due to study design, race, and the age range of the participants.

Recently, the average age of menarche in the Korean adolescent population has decreased [20, 21]. The current study highlights the importance of early menarche regarding diabetes and MetS risk and identifies new high-risk female subgroups for clinicians to monitor and treat to reduce the burden of diabetes.

The exact mechanism regarding the age of menarche and metabolic disease is not fully understood. Childhood hyperinsulinemia may cause early menarche, type 2 diabetes, and MetS. Hyperinsulinemia is usually caused by childhood obesity and may contribute to early sexual maturation [22]. This long-term insulin resistance increases blood glucose levels and may act as a risk factor for type 2 diabetes and MetS. Another hypothesis is that early menarche is associated with elevated C-reactive protein levels (CRP) and fasting and postprandial blood glucose levels. Elevated CRP levels are reportedly associated with an increased risk of type 2 diabetes and may play an indirect role in insulin resistance [23]. Further studies are needed to explain the underlying mechanisms mediating early menarche and metabolic disease.

A strength of this study is that it identifies the characteristics of postmenopausal Korean women using representative data. However, there are also some limitations to our study. First, there may be an information bias because we relied on memory to report menarche age. This may be inaccurate, especially in an elderly population. However, a prospective study has demonstrated a high correlation between the age of menarche recalled in adulthood and the actual age of menarche [24]. Second, since we didn't check the serum insulin and c-peptide levels, there is no information on hyperinsulinemia, the pathogenesis of diabetes, and MetS. Third,

this is a cross-sectional study, and we cannot establish a causal relationship between early menarche and diabetes. However, we can declare an inverse causal relationship between the age of menarche and diabetes as menarche began before the diagnosis of diabetes.

In conclusion, the risk of diabetes was 1.43 times higher in postmenopausal Korean women with early menarche. Early menarche is an important determinant of future diabetes, MetS, and related morbidity. Our results suggest that assessing the age of menarche may be helpful in diabetes risk stratification and early interventions for postmenopausal women. Further prospective studies are needed to clarify the role of menarche age on diabetes and MetS.

## Author Contributions

**Conceptualization:** Yeon-Ah Sung, Young Sun Hong, Do Kyeong Song, So-hyeon Hong, Hyejin Lee.

**Formal analysis:** Hyein Jung, Hyejin Lee.

**Investigation:** Hyein Jung, Yeon-Ah Sung, Young Sun Hong, Do Kyeong Song, So-hyeon Hong, Hyejin Lee.

**Methodology:** Yeon-Ah Sung, Young Sun Hong, Do Kyeong Song, So-hyeon Hong, Hyejin Lee.

**Project administration:** Young Sun Hong, Do Kyeong Song, So-hyeon Hong, Hyejin Lee.

**Supervision:** Yeon-Ah Sung, Young Sun Hong, Do Kyeong Song, So-hyeon Hong, Hyejin Lee.

**Validation:** Hyein Jung, Yeon-Ah Sung, Young Sun Hong, Do Kyeong Song, So-hyeon Hong, Hyejin Lee.

**Writing – original draft:** Hyein Jung.

**Writing – review & editing:** Hyein Jung, Hyejin Lee.

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
