## [Editor Report · Decision Letter 0]

12 Aug 2022

PONE-D-22-21289Relationship between age at menarche and metabolic diseases in Korean postmenopausal women: The Korea National Health and Nutrition Examination Survey 2016-2018PLOS ONE

Dear Dr. Lee,

Thank you for submitting your manuscript to PLOS ONE. After careful consideration, we feel that it has merit but does not fully meet PLOS ONE’s publication criteria as it currently stands. Therefore, we invite you to submit a revised version of the manuscript that addresses the points raised during the review process.

ACADEMIC EDITOR:I performed the first reading on your work, it seems to be very promising, however, so that we can proceed with the peer analysis, there is a need to adjust the format of the citations, and to review the best formatting for table 1.

We look forward to receiving your revised manuscript.

Kind regards,

Leonardo Costa Pereira, Doctor

Academic Editor

PLOS ONE
---

## [Author Response · Author response to Decision Letter 0]

19 Aug 2022

PONE-D-22-21289

Relationship between age at menarche and metabolic diseases in Korean postmenopausal women: The Korea National Health and Nutrition Examination Survey 2016-2018

PLOS ONE

Dear Dr. Lee,

Thank you for submitting your manuscript to PLOS ONE. After careful consideration, we feel that it has merit but does not fully meet PLOS ONE’s publication criteria as it currently stands. Therefore, we invite you to submit a revised version of the manuscript that addresses the points raised during the review process.

ACADEMIC EDITOR:

I performed the first reading on your work, it seems to be very promising, however, so that we can proceed with the peer analysis, there is a need to adjust the format of the citations, and to review the best formatting for table 1.

-> Thank you for your kind advice. As you recommended, we change the format of the citations and table.

---

## [Editor Report · Decision Letter 1]

22 Aug 2022

PONE-D-22-21289R1Relationship between age at menarche and metabolic diseases in Korean postmenopausal women: The Korea National Health and Nutrition Examination Survey 2016-2018PLOS ONE

Dear Dr. Lee,

Thank you for submitting your manuscript to PLOS ONE. After careful consideration, we feel that it has merit but does not fully meet PLOS ONE’s publication criteria as it currently stands. Therefore, we invite you to submit a revised version of the manuscript that addresses the points raised during the review process.

ACADEMIC EDITOR: Glad you got back in time with some of the requested changes. However, the paper lacks attention, with regard to the format of the citations, where I invite you to review them, following the regulations by the International Committee of Medical Journal Editors (ICMJE).

We look forward to receiving your revised manuscript.

Kind regards,

Leonardo Costa Pereira, Doctor

Academic Editor

PLOS ONE
---

## [Author Response · Author response to Decision Letter 1]

29 Aug 2022

PONE-D-22-21289

Relationship between age at menarche and metabolic diseases in Korean postmenopausal women: The Korea National Health and Nutrition Examination Survey 2016-2018

PLOS ONE

Dear Dr. Lee,

Thank you for submitting your manuscript to PLOS ONE. After careful consideration, we feel that it has merit but does not fully meet PLOS ONE’s publication criteria as it currently stands. Therefore, we invite you to submit a revised version of the manuscript that addresses the points raised during the review process.

ACADEMIC EDITOR:

I performed the first reading on your work, it seems to be very promising, however, so that we can proceed with the peer analysis, there is a need to adjust the format of the citations, and to review the best formatting for table 1.

-> Thank you for your kind advice. As you recommended, we changed the format of the citations and table. 

PONE-D-22-21289R1

Relationship between age at menarche and metabolic diseases in Korean postmenopausal women: The Korea National Health and Nutrition Examination Survey 2016-2018

PLOS ONE

Dear Dr. Lee,

Thank you for submitting your manuscript to PLOS ONE. After careful consideration, we feel that it has merit but does not fully meet PLOS ONE’s publication criteria as it currently stands. Therefore, we invite you to submit a revised version of the manuscript that addresses the points raised during the review process.

ACADEMIC EDITOR: Glad you got back in time with some of the requested changes. However, the paper lacks attention, with regard to the format of the citations, where I invite you to review them, following the regulations by the International Committee of Medical Journal Editors (ICMJE).

-> We apologize for not clarifying this point. Reference number 16 was changed from the website to a more accurate journal. Also, we remove the Endnote code and convert to plain text.

---

## [Editor Report · Decision Letter 2]

12 Jan 2023

Relationship between age at menarche and metabolic diseases in Korean postmenopausal women: The Korea National Health and Nutrition Examination Survey 2016-2018

PONE-D-22-21289R2

Dear Dr. Lee,

We’re pleased to inform you that your manuscript has been judged scientifically suitable for publication and will be formally accepted for publication once it meets all outstanding technical requirements.

Kind regards,

Tatsuo Shimosawa, M.D., Ph.D.

Academic Editor

PLOS ONE

Additional Editor Comments (optional):

I, as the alternative academic editor, evaluated your responses and revised manuscript to find it is acceptable in the current form. I apologize the delay of responses.
---

## [Editor Report · Acceptance letter]

16 Jan 2023

PONE-D-22-21289R2 

Relationship between age at menarche and metabolic diseases in Korean postmenopausal women: The Korea National Health and Nutrition Examination Survey 2016-2018 

Dear Dr. Lee:

I'm pleased to inform you that your manuscript has been deemed suitable for publication in PLOS ONE. Congratulations! Your manuscript is now with our production department. 

Kind regards, 

on behalf of

Prof. Tatsuo Shimosawa 

Academic Editor

PLOS ONE